# Longitudinal Changes in the Body Composition of Non-Institutionalized Spanish Older Adults after 8 Years of Follow-Up: The Effects of Sex, Age, and Organized Physical Activity

**DOI:** 10.3390/nu16020298

**Published:** 2024-01-18

**Authors:** Ana Moradell, Alba Gomez-Cabello, Asier Mañas, Eva Gesteiro, Jorge Pérez-Gómez, Marcela González-Gross, Jose Antonio Casajús, Ignacio Ara, Germán Vicente-Rodríguez

**Affiliations:** 1EXER-GENUD (Growth, Exercise, Nutrition and Development) Research Group, Universidad de Zaragoza, 50009 Zaragoza, Spain; amoradell@unizar.es (A.M.); agomez@unizar.es (A.G.-C.); joseant@unizar.es (J.A.C.); 2Department of Animal Production and Food Technology, Faculty of Health and Sport Science FCSD, University of Zaragoza, 50009 Zaragoza, Spain; 3Defense University Center, 50090 Zaragoza, Spain; 4Centro de Investigación Biomédica en Red de Fisiopatología de la Obesidad y Nutrición (CIBEROBN), Instituto de Salud Carlos III, 28040 Madrid, Spain; 5Agrifood Research and Technology Centre of Aragón-IA2, CITA-Universidad de Zaragoza, 50009 Zaragoza, Spain; 6GENUD Toledo Research Group, Faculty of Sports Sciences, Universidad de Castilla-La Mancha, 13003 Toledo, Spain; asier.manas@uclm.es (A.M.); ignacio.ara@uclm.es (I.A.); 7CIBER de Fragilidad y Envejecimiento Saludable (CIBERFES), Instituto de Salud Carlos III, 28029 Madrid, Spain; 8Instituto de Investigación Sanitaria de Castilla-La Mancha (IDISCAM), Junta de Comunidades de Castilla-La Mancha (JCCM), 45071 Toleldo, Spain; 9Center UCM-ISCIII for Human Evolution and Behavior, 28040 Madrid, Spain; 10Faculty of Education, Complutense University of Madrid, 28040 Madrid, Spain; 11ImFINE Research Group, Universidad Politécnica de Madrid, 28040 Madrid, Spain; egesteiro@upm.es (E.G.); marcela.gonzalez.gross@upm.es (M.G.-G.); 12HEME Research Group, University of Extremadura, 10003 Cáceres, Spain; jorgepg100@unex.es; 13Department of Physiatry and Nursing, Faculty of Medicine, University of Zaragoza, 50009 Zaragoza, Spain; 14Department of Physiatry and Nursing, Faculty of Health and Sport Sciences, University of Zaragoza, 50009 Zaragoza, Spain

**Keywords:** exercise, fat mass, muscle mass, obesity, sarcopenia

## Abstract

Aging leads to physiological changes affecting body composition, mediated by lifestyle. However, the effectiveness of organized physical activities (OPAs) in attenuating or delaying these age-related transformations remains an area of limited understanding. The primary objectives of this study were threefold: (I) to comprehensively assess the evolution of body composition in a cohort of Spanish older adults over an 8-year period; (II) to compare this evolution in the different age groups; and (III) to investigate the influence of active engagement in OPAs on these age-related changes. From a sample of 3136 Spanish older adults recruited in 2008, 651 agreed to participate in the 8-year follow-up. Anthropometric and bioelectrical impedance data were included for 507 females (70.3 ± 4.4 years) and 144 males (77.8 ± 4.5 years). Age groups were categorized as follows: youngest (65–69 years), mid (70–74 years), and oldest (≥75 years). The engagement in OPA was recorded before and after the follow-up. A repeated measures ANOVA was performed to evaluate the 8-year changes. Males increased in hip (98.1 ± 9.3 vs. 101.5 ± 10.2 cm) and waist circumferences (101.2 ± 6.6 vs. 103.2 ± 6.1 cm), specifically in the youngest group (*p* < 0.05). Females decreased in weight (67.6 ± 10.0 vs. 66.6 ± 10.5 kg) and fat mass percentage (39.3 ± 5 vs. 38.8 ± 5.4%) and increased in hip circumference (104.4 ± 9.0 vs. 106.5 ± 9.7 cm); these effects were the most remarkable in the oldest group (all *p* < 0.05). OPA engagement seemed to slow down fat-free mass loses in males, but not in females (grouped by time, *p* < 0.05). Body composition changes caused by aging seem to happen earlier in males than in females. Moreover, participating in OPAs does not prevent fat-free mass due to aging.

## 1. Introduction

The older population is continuing to grow, now faster than ever, mainly due to an increase in life expectancy in recent decades. As in most developed countries, in Spain, people above 65 years constitute 19.1% of the total population, and it is expected that in 2068, they will make up 29.4% [1].

Aging is characterized by physiological changes resulting in a progressive and generalized impairment in several bodily functions, an increased vulnerability to environmental challenges, and a growing risk of disease and risk of death [2,3]. Regarding changes in body composition, it has been observed that during the fourth decade of life, the peak of muscle mass starts to decrease [4,5]; there is an increase in and redistribution of fat mass (FM), with a subsequent accumulation in the visceral area [6]. Moreover, bone mass and density also suffer a decrease until elderhood [7]. All these changes contribute to the development of three significant physiopathologies—sarcopenia, obesity, and osteoporosis—which could develop simultaneously, increasing the risk of experiencing frailty and dependence during the aging process [8].

These health-related problems associated with aging impose high costs on social and health systems and services for older adults, which could be reduced through physical activity (PA) [9]. In this regard, the benefits of PA on overall health have been previously demonstrated in this population [10]. Specifically, PA has been associated with a decrease in mortality and cardiovascular diseases [11,12], constituting one of the most important findings. Furthermore, exercise can improve body composition by increasing skeletal muscle mass and reducing FM, which may contribute to higher independence in the ageing [13]. The most common PA in older adults is walking [14]; however, the intensity of walking activities alone is not enough to induce physiological changes leading to body composition changes, and it should be combined with strength exercises [15]. City halls, sport centres, and institutions also offer organized physical activities (OPAs) for older adults, constituting the second main PA among this population. These types of OPAs commonly include a variety of exercises that combine flexibility, balance, strength, and endurance, which could increase physical fitness. It is thought that OPAs may not stimulate this population enough to achieve the potential benefits of exercise, as they lack a planned progression in loads and increases in the demands of training. However, the effect of these OPAs on changes in body composition is still unknown. Nevertheless, OPAs should be studied in depth to determine if they can be considered as a measure for preventing body composition changes caused by aging, and to identify the characteristics that make these activities effective.

Therefore, the main aims of this paper are (1) to describe changes in body composition (fat mass index, fat-free mass index, waist and hip circumferences) during an 8-year follow-up in a large sample of Spanish seniors aged ≥65 who were engaged in OPAs; (2) to compare this evolution in different sex and age groups (≥65 to 69 vs. 70 to 74 vs. ≥75 years old), in order to test whether there is a critical decline at a specific age group; and (3) to evaluate the effects of engagement in OPA on changes in body composition.

## 2. Materials and Methods

### 2.1. Sample of the Study

This study was carried out within the framework of the longitudinal elderly EXERNET study (EXERNET-Elder 3.0), a multi-centric study performed between 2008 (baseline) and 2016 (follow-up) on a representative sample of 3136 Spanish older adults. This population was selected through a multistep, simple random sampling from the locations of six different regions from Spain: Aragón, Castilla-La Mancha, Castilla y León, Madrid, Extremadura, and Canarias, in order to ensure the geographical and cultural diversity of the sample [16]. Initial inclusion criteria comprised individuals aged 65 years and above who were independent or able to take care of themselves and not living in nursing homes and those not suffering from cancer and/or dementia. The recruitment of the initial sample took place in older adult centres where PA were developed for this population. Thus, most of the participants could be considered active.

For the baseline evaluation, the region of Canarias was unable to participate, resulting in the loss of 400 subjects. A total of 740 participants agreed to continue in the follow-up measurements. Considering that 260 deaths were registered from 2008 to 2016, other reported causes included changes of residence, failure to answer the phone, change to another city, becoming dependent, or declining to participate. From the initial 740, 17 participants did not attend the day when body composition was assessed, and 61 were not engaged in regular OPA during the baseline, which was established as an inclusion criterion for this study. Therefore, the data for the 651 participants (144 males) who completed anthropometric and body composition measurements are described in the present manuscript.

Personal, sociodemographic, PA, and sedentarism (≥4 h sitting/day) information was collected through a structured and validated questionnaire [16], followed by a blood test and anthropometrics, bone mass and structure tests, and fitness assessments. Written informed consent was obtained from all the included participants. The protocol of the study was performed in accordance with the Helsinki Declaration of 1961 (revised in Washington 2002) and in Fortaleza (2013) [17], and it was approved by the Clinical Research Ethics Committee of Aragón (18/2008) for the baseline and by the Hospital Universitario Fundación de Alcorcón (16/50) for the follow-up in 2016.

### 2.2. Anthropometric Measurements

A portable stadiometer with 2.10 m maximum capacity and 1 mm error margin (Seca 711, Hamburgo, Germany) was used to measure height. Subjects stood with their scapula, buttocks, and heels resting against a wall; the neck was held in a natural non-stretched position; the heels were touching each other with the toe tips spread to form a 45° angle; and the head was held straight with the inferior orbital border in the same horizontal plane as the external auditory tube (Frankfort’s plane). Waist circumference was taken at the narrowest point between the lower costal border and the iliac crest, and hip circumference was taken at the level of the greatest posterior protuberance of the buttocks. All anthropometric measurements followed the standards of the International Society for the Advancement of Kinanthropometry (ISAK) [18]. Training workshops were organized to harmonize the assessment of anthropometric measurements before starting both evaluations of the study [19].

### 2.3. Weight, Fat Mass, and Fat-Free Mass

A body composition analyser with a 200 kg maximum capacity and a ±50 g error margin (TANITA BC-418MA, Tanita Corp., Tokyo, Japan) was used to measure body weight (kg) and to estimate whole-body total FM, the percentage of body fat mass (FM%), and fat-free mass (FFM). Individuals had to remove shoes and heavy clothes before weighing. Body mass index (BMI) was calculated by dividing weight (kg) by squared height (m^2^). The fat mass index (FMI) and fat-free mass index (FFMI) were calculated dividing by squared height (m^2^).

### 2.4. Physical Activity and Sedentary Behaviour

Information about lifestyles was collected by researchers through an interviewed validated questionnaire [18]. Participation in OPA was recorded through a yes–no question: “Are you participating in organized physical activity?”, and an ad hoc question about the duration of these activities: “How much time do you spend on it per week?”. At the end of the follow-up, those who continued doing OPA were categorized as ALWAYS OPA to compare with those who STOPPED OPA participation. Participants were also asked to specify the type of activity: supervised gym for keeping fit, dancing, yoga, Pilates, etc. Sedentary time and walking time were recorded through the following question: “How many hours do you usually spend sitting per day?”, and “How many hours do you usually spend walking per day?”. The sedentary time question involved all activities in which the person was sitting, such as watching television, reading, sewing, etc. The cut-off points to define this sedentary behaviour were <4 h/day for non-sedentary individuals and ≥4 h/day for sedentary individuals, based on receiver operating characteristics (ROC) curves produced with the same sample and reported in a previous study [20]. All this information was collected in baseline and follow-up evaluations.

### 2.5. Mediterranean Diet Adherence

The Mediterranean Diet Adherence Screener was used to evaluate the dietary patterns of the participants. It is a 14-item questionnaire designed to assess the regularity and quantity of consumption of key components and dietary habits associated with the Mediterranean diet. It was created by 14 yes–no questions about the habitual frequency of various food items related to the Mediterranean diet [21]. The higher the score, the greater the adherence. This questionnaire was only obtained in the follow-up. Among others, the questionnaire scores the intake of one glass of wine per day, and the daily use of olive oil, fruit, and vegetables as positive.

### 2.6. Statistical Analysis

The Statistical Package for the Social Sciences (SPSS) v. 20.0 for Windows (SPSS, Inc., Chicago, IL, USA) was used to analyse the data. All the analyses were performed with the sample divided by sex. The normality of the variables was checked with a Kolgomorov–Smirnoff test. The central limit theorem was applied to assume the normality of the non-normal distributed variables [22]. Dependent sample *t*-tests were performed to evaluate differences between baseline and follow-up evaluation for descriptive continuous variables, while chi-square tests were developed to evaluate categorical data. The mean and standard deviation or the number of participants and the percentage from the total sample are reported, respectively.

Participants were grouped according to age into three groups: 65–69.9 years: youngest (*n* = 72 males and 222 females), 70–74.9 years: mid (*n* = 45 males and 173 females), and ≥75 years: oldest (*n* = 27 males and 112 females). In order to evaluate differences in body composition variables between the two evaluations and to test differences between age groups, a repeated measures analysis of variance (ANOVA) was developed. In those showing group–time interaction, further post hoc contrasts were performed to study between which age groups there were statistically significant differences.

To analyse differences in the evolution of body composition considering OPA engagement upon follow-up, two groups were created (always OPA and stopped OPA), and additional repeated measures analyses of covariance (ANCOVA), adjusting by age, were performed.

These analyses were repeated, adjusting separately by income, OPA hours/week, walking time/day, Mediterranean diet adherence score, and wine consumption, in order to test if there was an interaction with these confounders.

Effect size statistics using partial eta squared (ηp^2^) for repeated measures are reported. The effect sizes were considered small (0.01–<0.06), medium (0.06–<0.14), or large (>0.14) [23]. Statistical significance was set at *p* < 0.05.

## 3. Results

Descriptive characteristics of participants are shown in Table 1.

The proportion of participants who participate in more than one activity was 23.5%. Figure 1a,b represent the most common OPAs reported by participants.

### 3.1. Body Composition Changes during the 8-Year Follow Up, Stratified by Sex

Height, FM%, and waist circumference showed group–time interactions between males and females (all *p* < 0.05; Table 2).

Males showed a significant decline in height (medium effect), while waist and hip circumferences increased with time (small effect) (all *p* < 0.05; Table 2).

In females, there were significant declines in height (large effect) and in weight and FM%, with a small effect. Hip circumference showed an increase with a medium effect (all *p* < 0.05; Table 2).

### 3.2. Body Composition Changes during the 8-Year Period Stratified by Sex and According to Age Group

Table 3 shows body composition changes by age and sex groups.

No group–time interactions were observed in males (*p* > 0.05). Despite this fact, some statistically significant changes were observed within groups. Specifically, the young group showed a decline in height with a large effect, as well as significant increases in FM% (with a small effect). Waist and hip circumference also increased with a medium effect. The mid group showed significant declines in height (with a large effect) and in waist circumference (with a medium effect). Meanwhile, the oldest group only showed changes in height and hip circumferences with large and medium effects, respectively. Further contrast between those groups showed differences in height between the youngest and mid groups, differences in FM% between mid and both the youngest and oldest groups, differences in FMI between the mid and oldest groups, and in hip circumference between the youngest and mid groups (*p* < 0.05).

For females, group–time interactions were observed in height, FM%, and waist circumference. Changes within groups between evaluations were found for the youngest group, which showed a decrease in height and increases in waist circumference (small effect) and hip circumference (medium effect) (all *p* < 0.05). The mid group showed a decline in height, (large effect) and weight (small effect), while their hip circumference increased (small effect) (all *p* < 0.05). In the oldest group, declines were found for height (large effect), weight (small effect), FMI, and FM% (both with a small effect), while hip circumference increased (small effect) (all *p* < 0.05). Further contrast showed differences between the three groups for the mean change in height in the youngest group compared with both the mid and oldest groups, and also when comparing the mid group with the oldest group (all *p* < 0.05).

### 3.3. Body Composition Changes Stratified by Sex According to Changes in OPA

A total of 29 males had given up OPA in the follow-up. Body composition changes within and between groups in the always OPA and the stopped OPA groups are presented in Table 4.

Except for weight, the always OPA group had lower baseline and follow-up means compared to the stopped OPA group, all with a small effect size.

The always OPA group showed significant changes from baseline to follow-up for height, waist and hip circumference, with a small effect (all *p* < 0.05). The stopped OPA group showed changes in height, weight (both with a small effect), and FFMI with a medium effect size (all *p* < 0.05). Group–time interactions were found in FFMI (ηp^2^ = 0.050), with the stopped OPA group having a more remarkable decrease in FFMI than the always OPA group (GxT < 0.05).

In females, a total of 101 had given up OPA in the follow-up. Changes within the always OPA and stopped OPA groups are shown in Table 4. As happened in males, except for weight, the always OPA group had lower baseline and follow-up means compared to the stopped OPA group. Baseline significant differences were observed in weight, BMI, FFMI, FMI, FM%, and hip circumference. In the follow-up, differences were observed in weight, BMI, FMI, FFMI, and waist and hip circumference, all with a small effect size.

Differences between baseline and follow-up were found for the always OPA group in height, weight, %FM, and hip circumference (all *p* < 0.05). The stopped OPA group showed significant decreases in height, FM, and an increase in waist circumference and hip circumference. However, changes between groups were not statistically significantly different (GxT *p* > 0.05).

Only income presents interactions with time for waist circumference and fat mass percentage (*p* < 0.05). However, it did not affect the significant differences found. Same statistically significant differences were observed when analyses were adjusted by OPA hours, walking hours, or adherence to a Mediterranean dietary pattern and wine intake.

## 4. Discussion

The relevance and importance of this study lies in its 8-year longitudinal design. The main findings of the present study were that (1) both genders showed a decline in height and a redistribution of adiposity during ageing; (2) in females, there was a decrease in weight after the age of 70, which simultaneously continues with a decrease in FMI and FFMI; and (3) there was a decrease in FFMI only in those males who stopped participating in OPA, but otherwise, it did not appear to have an effect.

Several studies have previously documented changes in body composition during aging; however, those with larger sample sizes were conducted in Asian populations [24,25,26] and had a shorter duration [27,28] or smaller samples at these ages [29]. As in our study, height reductions have been reported; this might be due to vertebral compression that has been related to bone fragility [30]. For weight, males did not seem to suffer significant variations, while females showed a decrease, specifically, in the mid and oldest groups, in concordance with other populations [26,31]. On the contrary, different patterns in weight among people aged 70–79 have been reported: those who lose weight, those who gain it, and those whose weight is stable [27]. This could explain why males did not show the same results. Independently of the changes in weight, BMI seemed not to change during this time. It also did not happen in the large sample of the “*Health, Aging and body composition study*” [28]. Nevertheless, other authors do not recommend using BMI in this population, because this parameter can be altered by the loss of muscle mass that they suffer [32]. In line and in combination with this, as changes in height occur during aging, they could also affect BMI values and changes, making it difficult to elucidate whether their differences or their absences are really due to tissue-related changes during aging.

In this regard, only females, when analysed together, showed a decline in FFMI that seemed to start in the oldest group (70–75 y). Previous authors have described these same trajectories in studies of Asian populations of more than three hundred participants with a 5-year follow-up, as measured by DXA [24]. In contrast, another study of a large sample of community-dwelling Japanese people aged 40–79 years suggested that lean arm and leg mass measured by DXA decreased markedly in men in their 70s, while it had already decreased in females in their 40s [25]. Nevertheless, it is difficult to compare these data with our Spanish sample, as both studies were carried with Japanese populations, and culture and ethnicity could influence changes. It has been found that compared with Caucasians, Asians present higher values of FM and lower values of skeletal muscle mass [33,34].

Regarding FM, a previous study with a larger male population has described an increase from age 20 upwards that levelled off at approximately 80 years, as emerged from our longitudinal results [29]. For Japanese women, an increase in FM was described until their fifties, after which they seem to maintain it [25], while in our sample of women, FM started to decrease in the oldest group. Our results suggest that in these advanced ages, a catabolic mechanism is activated. Studies with larger samples of older adults above 80 will also allow us to describe in detail what really happens in this very elderly population. In addition, regarding the expected redistribution of fat, our males showed an increment in hip and waist circumference, while females only increased in hip circumference. Likewise, in this other study, waist-to-hip ratio increased in men and women according to age group, specifically in those aged 65–74 years, indicating increased abdominal fat [31]. These increments in waist and hip measurements indicate a higher risk of mortality [35], so they require special attention when designing strategies to improve health in this population.

It could be thought that participating in active behaviour (being engaged in OPA for instance) can attenuate body composition deterioration through aging. In this case, it is good news that no loss in FFM was found in those males engaged in OPA, as happened in those who stopped. Therefore, OPA seemed to be enough to stimulate the maintenance of muscle mass in males. For females, there was a loss in FM% in both groups, and an increase in hip and waist circumference were observed for those who stopped OPA, which could be related to a redistribution of adipose tissue and higher risk of diseases. These types of activities may not be stimulating enough to attenuate age changes, as more significant declines were observed in those who continue participating in OPA. It has been mentioned that these changes in females could be also related to a decrease in time spent engaging in OPA. Intensities and amount of activity within these OPA programs are unknown, and do not allow us to further elucidate the convenience of the physical stimulus. Additionally, people participating in OPA may have the idea that they are more protected against fat accumulation or redistribution, despite to low amounts of activity or low-intensity activity. However, these ideas should be specifically addressed in future research. These sex differences could be explained by the higher values of FFMI in males, which could mean they need a lighter stimulus than females to maintain FFMI; alternatively, the intensity at which males exercise during OPA may be higher. Again, new specific studies may bring some answers to these questions. Nevertheless, Genton et al. reported that PA could limit lean tissue loss in males, but not in females, using a shorter sample [31]. Another longitudinal study over 3 years of follow-up showed that FFM decreased while FM increased, without fluctuations in weight. Besides, PA as measured by questionnaires seemed not to stop this decline through aging [13]. Haughes et al. also observed FFM decreases independently of engagement in recreational sports activities. Similarly, these large samples, followed over 6 years, emphasized the idea that women did not show changes despite their level of PA (related to aerobic exercise) [36], and neither did males (in which different PA trajectories showed no differences in FFM, measured by DXA over 5–7 years) [37]. From these overall results, it seems clear that evaluating the intensity of OPA sessions should be considered in the future so that we can ensure that PA is used as an efficient tool for the promotion of healthy aging. In fact, further research should design and implement specific exercise programs to elucidate the optimal dose needed for enhancement of body composition variables at these ages.

Moreover, despite the fact that adjusting for Mediterranean diet and wine intake could influence our results due to their relationship with diseases such as metabolic syndrome [38] and mortality risk [39], they seemed not to affect the results.

Some strengths and limitations of this study should be highlighted. The present study evaluates 8 years of body composition trajectories in an older adult population. However, multiple losses in the follow-up happened. Moreover, the whole initial sample was active, so results could not be extrapolated to the general older population. Further research including more males is required to verify the present results. Including the measurement of the intensities of these activities would help to establish robust conclusions. Additionally, despite the answers about OPA and sedentary time being subjective, the questionnaire was validated. Finally, other strengths like harmonised assessments, well-instructed researchers, women-specific results (diminishing the gap in women’s health-related research), and sample size should be considered.

## 5. Conclusions

In females, aging changes were characterized by a loss in weight and FMI, with body composition deterioration being more pronounced in older females. In males, there was only an increase in waist and hip circumference when they were analyzed all together. However, males who discontinued OPA experienced a decrease in their FFMI, an effect not seen in females. These findings emphasize the need for tailored interventions to address age-related body composition changes in older adults, with consideration of gender-specific factors. Further research is warranted to deepen our understanding of the underlying mechanisms and to develop more effective strategies for mitigating the impact of aging on body composition and overall health.

## Figures and Tables

**Figure 1 nutrients-16-00298-f001:**
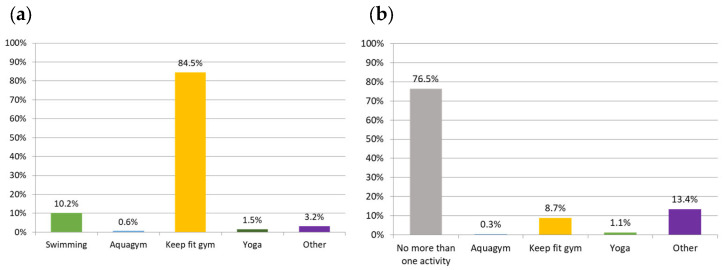
Types of organized physical activity performed as (**a**) a first activity and (**b**) a second OPA.

**Table 1 nutrients-16-00298-t001:** Descriptive characteristics of the sample.

		Males (*n* = 145)			Females (*n* = 506)		
		Baseline (2008)	Follow-Up (2016)	*p* Value	Baseline (2008)	Follow-Up (2016)	*p* Value
Age years		70.3 ± 4.4	77.8 ± 4.5	<0.001	70.7 ± 4.7	78.4 ± 4.7	<0.001
physical activity and sedentarism							
OPA	Yes (%)	145 (100.0)	116 (80.0)	<0.001	506 (100.0)	410 (79.8)	<0.001
	No (%)	0(0.0)	29 (20.0)		0(0.0)	104 (20.2)	
Weekly hours of OPA		3.9 ± 2.5	3.8 ± 3.2	0.913	3.5 ± 2.5	3.0 ± 1.5	<0.001
Sedentary	Yes (%)	43 (29.7)	68 (46.8)		236 (46.6)	154 (30.4)	<0.001
	No (%)	102 (70.3)	77 (53.2)		270 (53.4)	352 (69.6)	
Daily sitting	<1 h	1 (0.8)	1 (0.7)	0.007	11 (2.2)	3 (0.6)	<0.001
	1–2 h	13 (9.2)	8 (5.6)		53 (10.6)	50 (9.9)	
	2–3 h	44 (30.5)	33 (22.9)		160 (31.7)	104 (20.5)	
	3–4 h	476 (32.1)	34 (23.6)		127 (25.1)	113 (22.4)	
	4–5 h	20 (13.7)	37 (25.7)		81 (16.0)	119 (23.4)	
	>5 h	20 (13.7)	31 (21.5)		73 (14.4)	117 (23.2)	
Daily walking	<1 h	35 (24.1)	32 (22.2)	0.004	169 (33.4)	206 (40.7)	<0.001
	1–2 h	75 (51.7)	75 (51.7)		256 (50.6)	232 (45.8)	
	2–3 h	26 (17.9)	27 (18.3)		72 (14.2)	41 (8.1)	
	3–4 h	3 (2.1)	7 (5.0)		6 (1.2)	14 (2.8)	
	4–5 h	3 (2.1)	3 (2.2)		2 (0.4)	6 (1.2)	
	>5 h	-	1 (0.6)		2 (0.4)	7 (1.4)	

*p* < 0.05 for the paired t-samples and chi-square analysis. OPA: organized physical activity.

**Table 2 nutrients-16-00298-t002:** Evolution of the body composition variables.

	Males (*n* = 145)			Females (*n* = 506)			GxT *p* Value
	Baseline (2008)	Follow-Up (2016)	ηp^2^	Baseline (2008)	Follow-Up (2016)	ηp^2^	
Height (cm)	165.3 ± 5.9 ^†^	164.5 ± 5.9 ^†^	0.082 *	152.5 ± 5.8	151.4 ± 5.9	0.417 *	<0.001
Weight (kg)	78.2 ± 9.4 ^†^	77.6 ± 9.7 ^†^	0.003	67.6 ± 10.0	66.6 ± 10.5	0.032 *	0.347
BMI (kg/m^2^)	28.6 ± 3.2	28.6 ± 3.3	0.000	29.1 ± 4.1	29.1 ± 4.3	0.000	0.810
FFMI (kg/m^2^)	20.2 ± 1.7 ^†^	20.1 ± 1.8 ^†^	0.000	17.5 ± 1.6	17.6 ± 1.7	0.007	0.064
FMI (kg/m^2^)	8.3 ± 2.0 ^†^	8.5 ± 2.1 ^†^	0.001	11.6 ± 2.9	11.5 ± 3.1	0.007	0.062
FM%	28.8 ± 4.5 ^†^	29.3 ± 4.8 ^†^	0.004	39.3 ± 5.0	38.8 ± 5.4	0.016 *	0.005
Waist Cir (cm)	98.1 ± 9.3 ^†^	101.5 ± 10.2 ^†^	0.031 *	91.0 ± 12.2	91.7 ± 11.4	0.005	0.002
Hip Cir (cm)	101.2 ± 6.6 ^†^	103.1 ± 6.1 ^†^	0.026 *	104.4 ± 9.0	106.5 ± 9.7	0.114 *	0.554

* Statistically significant differences between baseline and follow-up, ^†^ statistically significant differences between sexes; GxT: group–time interaction. All statistically significant differences were set at *p* < 0.05. BMI: body mass index, FMI: fat mass index, FFMI: fat-free mass index, FM%: fat mass percentage, Cir: circumference.

**Table 3 nutrients-16-00298-t003:** Body composition stratified by age groups.

		Males				Females			
		Baseline (2008)	Follow-Up (2016)	ηp^2^	GxT *p* Value	Baseline (2008)	Follow-Up (2016)	ηp^2^	GxT *p* Value
Height (cm)	65–<70 y	166.3 ± 5.3	165.6 ± 5.2	0.235 *	0.570	153.7 ± 5.8	152.7 ± 5.9	0.214 *	0.022
	>70–<75 y	164.0 ± 6.7	163.1 ± 6.7	0.239 *		152.3 ± 5.6	151.1 ± 5.7	0.226 *	
	≥75 y	164.8 ± 5.5	163.9 ± 5.5	0.136 *		150.4 ± 5.2	149.0 ± 5.2	0.223 *	
Weight (kg)	65–<70 y	78.1 ± 10.3	77.8 ± 10.0	0.003	0.653	68.0 ± 9.9	67.5 ± 10.5	0.004	0.059
	>70–<75 y	79.0 ± 9.5	78.0 ± 9.7	0.020		67.6 ± 10.0	66.6 ± 10.2	0.015 *	
	≥75 y	76.0 ± 6.6	75.4 ± 9.3	0.005		65.6 ± 8.9	63.7 ± 9.5	0.046 *	
BMI (kg/m^2^)	65–<70 y	28.2 ± 3.3	28.3 ± 3.2	0.003	0.806	28.8 ± 4.0	29.0 ± 4.3	0.003	0.220
	>70–<75 y	29.4 ± 3.4	29.3 ± 3.7	0.001		29.2 ± 4.3	29.2 ± 4.4	0.000	
	≥75 y	27.9 ± 2.1	28.0 ± 2.9	0.000		29.0 ± 3.9	28.7 ± 3.9	0.003	
FFMI (kg/m^2^)	65–<70 y	20.1 ± 1.7	19.9 ± 1.7	0.012	0.778	17.4 ± 1.4	17.6 ± 1.6	0.005	0.187
	>70–<75 y	20.3 ± 1.7	20.1 ± 1.9	0.003		17.6 ± 1.9	17.6 ± 1.7	0.000	
	≥75 y	20.1 ± 1.5	20.1 ± 1.9	0.000		17.5 ± 1.4	17.8 ± 2.0	0.010 *	
FMI (kg/m^2^)	65–<70 y	8.0 ± 2.0	8.2 ± 1.9	0.019	0.358	11.4 ± 2.9	11.4 ± 3.2	0.001	0.020
	>70–<75 y	9.0 ± 2.2	9.2 ± 2.4	0.000		11.7 ± 2.7	11.7 ± 2.9	0.002	
	≥75 y	7.8 ± 1.3	7.6 ± 1.9	0.003		11.5 ± 2.6	10.9 ± 2.8	0.034 *	
FM%	65–<70 y	28.1 ± 4.7	29.1 ± 4.2	0.034 *	0.233	39.0 ± 5.1	38.7 ± 5.4	0.003	0.005
	>70–<75 y	30.6 ± 4.2	30.7 ± 4.9	0.001		39.4 ± 4.9	39.3 ± 5.0	0.000	
	≥75 y	27.8 ± 3.8	27.2 ± 5.4	0.004		39.3 ± 4.8	37.7 ± 5.6	0.039 *	
Waist Cir (cm)	65–<70 y	97.0 ± 9.1	100.6 ± 9.2	0.089 *	0.841	89.5 ± 11.5	91.0 ± 11.4	0.011 *	0.048
	>70–<75 y	99.9 ± 10.1	103.5 ± 12.7	0.061 *		91.5 ± 12.3	92.3 ± 11.8	0.003	
	≥75 y	97.4 ± 7.7	100.0 ± 7.8	0.019		92.9 ± 12.1	91.6 ± 10.0	0.006	
Hip Cir (cm)	65–<70 y	100.3 ± 7.0	102.4 ± 5.9	0.092 *	0.198	103.7 ± 8.6	106.9 ± 10.2	0.082 *	0.201
	>70–<75 y	103.2 ± 6.5	104.2 ± 6.8	0.014		104.8 ± 9.6	107.0 ± 9.6	0.051 *	
	≥75 y	100.0 ± 5.5	103.0 ± 5.3	0.074 *		104.1 ± 7.6	105.0 ± 8.2	0.013 *	

* Statistically significant differences within groups, *p* < 0.05. GxT: group–time. BMI: body mass index, FMI: fat mass index, FFMI: fat-free mass, FM%: fat mass percentage, Cir: circumference.

**Table 4 nutrients-16-00298-t004:** Body composition changes stratified by sex according to changes in OPA.

Males							
	Stopped OPA			Always OPA			GxT *p* Value
	Baseline (2008)	Follow-Up (2016)	ηp^2^	Baseline (2008)	Follow-Up (2016)	ηp^2^	
Height (cm)	165.4 ± 4.9	164.2 ± 5.0	0.151 *	165.4 ± 6.0	164.6 ± 6.0	0.401 *	0.513
Weight (kg)	82.0 ± 9.0 ^†^	80.2 ± 8.9	0.025 *	77.8 ± 9.4	77.2 ± 9.9	0.016	0.226
BMI (kg/m^2^)	30.0 ± 3.1 ^†^	30.0 ± 4.3	0.005	28.9 ± 4.0	28.9 ± 4.1	0.000	0.687
FFMI (kg/m^2^)	20.9 ± 2.1 ^†^	20.3 ± 2.1 ^†^	0.061 *	20.0 ± 1.5	20.0 ± 1.7	0.000	0.010
FMI (kg/m^2^)	8.8 ± 2.3	9.0 ± 1.9	0.469	8.1 ± 2.0	8.3 ± 2.1	0.285	0.848
FM%	29.4 ± 5.7	30.5 ± 4.4	0.005	28.7 ± 4.2	29.0 ± 4.2	0.017	0.282
Waist Cir (cm)	97.7 ± 9.1 ^†^	105.4 ± 14.3 ^†^	0.036 *	101.9 ± 9.4	105.4 ± 14.3	0.114 *	0.714
Hip Cir (cm)	103.8 ± 7.7 ^†^	104.6 ± 6.6	0.004	101.0 ± 6.2	103.0 ± 6.2	0.139 *	0.176
**Females**							
	**Stopped OPA**			**Always OPA**			**GxT** ***p* Value**
	**Baseline** **(2008)**	**Follow-Up** **(2016)**	**ηp^2^**	**Baseline** **(2008)**	**Follow-Up** **(2016)**	**ηp^2^**	
Height (cm)	152.0 ± 5.4	150.6 ± 5.7	0.175 *	152.8 ± 5.8	151.6 ± 5.9	0.394 *	0.171
Weight (kg)	69.7 ± 11.6 ^†^	68.6 ± 13.2 ^†^	0.009	67.3 ± 9.5	66.5 ± 9.8	0.026 *	0.925
BMI (kg/m^2^)	30.1 ± 4.3 ^†^	30.2 ± 5.2 ^†^	0.001	28.9 ± 4.0	28.9 ± 4.1	0.000	0.343
FFMI (kg/m^2^)	17.9 ± 1.6 ^†^	18.1 ± 1.9 ^†^	0.004	17.5 ± 1.6	17.6 ± 1.6	0.003	0.122
FMI (kg/m^2^)	12.4 ± 3.0 ^†^	11.3 ± 2.9 ^†^	0.001	11.5 ± 2.9	11.3 ± 2.9	0.007	0.903
FM%	40.5 ± 5.1	39.6 ± 6.7	0.010 *	39.1 ± 4.9	38.7 ± 5.1	0.012 *	0.320
Waist Cir (cm)	93.2 ± 13.6	95.2 ± 14.8 ^†^	0.003	90.4 ± 11.7	91.0 ± 10.3	0.011 *	0.120
Hip Cir (cm)	106.6 ± 10.5 ^†^	103.9 ± 8.4 ^†^	0.038 *	109.0 ± 8.4	106.2 ± 9.1	0.115 *	0.589

* Statistically significant differences between baseline and follow-up, ^†^ statistically significant differences between OPA groups, GxT: Group–time interaction. All statistically significant differences were set at *p* < 0.05. BMI: body mass index, FMI: fat mass index, FFMI: fat-free mass index, FM%: fat mass percentage, Cir: circumference.

## Data Availability

The data presented in this study are available on request from the corresponding author. The data are not publicly available due to privacy.

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
