# Peer review of "Longitudinal Changes in the Body Composition of Non-Institutionalized Spanish Older Adults after 8 Years of Follow-Up: The Effects of Sex, Age, and Organized Physical Activity"

_nutrients, 2024, doi:10.3390/nu16020298_

Round 1

Reviewer 1 Report

Comments and Suggestions for Authors

The article addresses themes that could be intriguing for Nutrients readers, after revision, particularly considering the increasing average age and current emphasis on lifestyle as a potential influencer of improved quality of life.

This study's notable strengths lie in its extensive patient follow-up and the inclusion of a substantial population as the subject of examination.

Major comments

Please remove all the quote that are in only in Spanish… as number 1and 16

The population involved in the study is numerically significant; however, there is a notable numerical difference between male and female participants, known to possess distinct physiological characteristics. Regarding you previous paper (quote number 22)  from which this population is derived, the Male/Female ratio was really different 6/4. Why now the male are less than the female (mow male/female ratio Is 1/5)? Could you provide some information and explanation?

Are the difference in age groups comparable with Spanish population?

Additionally, exploring the dietary patterns among the participants would have been valuable, given the crucial role of diet in influencing changes in body composition. This insight could have enhanced the identification of correlations between dietary habits and the observed physical variations within the study sample. Moreover, Spanish have a Mediterranean diet that can help to reduce age effect on body. This point need to be reported in the result section or addressed as limit in the discussion.

Could you add some information about alcohol intake?

Page 8 section “Body composition changes stratifying by sex according to changes in OPA” is really hard to read, please improve it reporting the majority of these results in a table that can help the reader.

Considering citation [5] at line 59, it would be advisable to cite a more recent study (including a systematic review) due to the numerous emerging studies that examine the correlation between sarcopenia and aging.

Please is important to include a citation at line 316 “Several studies have previously documented changes in body composition during aging; however, those with larger sample sizes were conducted in Asian populations or had a shorter duration.”

Minor comments a major English revision is required, following some examples:

L 34 – Please only use the abbrevation “OPA”

L 51 Please consider making a grammar revision: “The older population continues to grow”

L61: Please consider making a revision of “All these changes lead to suffer three important physio pathologies which also increase the risk of suffering frailty  and dependence during the aging process: sarcopenia, obesity and osteoporosis, which could be developed simultaneously”

I suggest you to use “All these changes contribute to the development of three significant physiopathologies,which […]”

L 75 - Please only use the abbrevation “OPA”

L 83-84: “which are the characteristics that makes effective these activities.” Please correct with: “characteristics that make these activitis effective.”

L 102: please remove activities after PA, as it is already present in the abbreviation PA

L 103: please remove as in the sentence: “Thus, most of the participants could be considered as active.”

L 105: please use the past tense: “agreed

L 148: please use the present tense: “spend” in the question: “How much time do you spent in them per week?”

L 152: please use “were” in the sentence: “Sedentary and walking time was recorded “

L 153: please use “walk” or “spend walking” in the question: “How many hours do you usually walking per day?”.

L225: please use “increase” instead of “increased”.

L 245: “in the Youngest group”

L 308: please use “lies” instead of “lie”

L 309: please add “the” before “main findings”

L 331: please use “continues” instead of “continue”

L 311: please use “there was” instead of “there is”

L 312: please revise the sentence “while in female seems not to affect.” I think it’s better using: “…while in females, it appears not to have an effect."

L 325: please make a grammar revision of this sentence: “In line, and combination with this, as changes in height occur during ageing, it could be affecting also BMI values and changings making it difficult to elucidate whether the differences or the absence of them are really due tissue-related changes during ageing”.

L 339: please add “a” before “previous study”

L 339: please consider to use “male” instead of “men”

L 341: please consider making a revision of “For Japanese women, it was described an increase in FM till their fifties and then, they seem to maintain it”. I suggest: “For Japanese women an increase in FM until their fifties was described, after which they seemed to maintain it…”

L 349: please consider making a revision of “Such increases in waist and hip indicates a higher risk of mortality …”  I suggest you to write “ These increments in waist and hip measurements indicate …”

L 367: please add “a” before “lighter”.

L 394: please use the past tense “there was” instead of “there is”

L 395-396: please consider making a revision of this sentence to improve clarity: “However, males who stopped OPA decrease their FFMI happened in males while it has no effect in females”. I suggest you to write: “However, males who discontinued OPA experienced a decrease in their FFMI, an effect not seen in females.”

Comments on the Quality of English Language

The article addresses themes that could be intriguing for Nutrients readers, after revision, particularly considering the increasing average age and current emphasis on lifestyle as a potential influencer of improved quality of life.

This study's notable strengths lie in its extensive patient follow-up and the inclusion of a substantial population as the subject of examination.

Major comments

Please remove all the quote that are in only in Spanish… as number 1and 16

The population involved in the study is numerically significant; however, there is a notable numerical difference between male and female participants, known to possess distinct physiological characteristics. Regarding you previous paper (quote number 22)  from which this population is derived, the Male/Female ratio was really different 6/4. Why now the male are less than the female (mow male/female ratio Is 1/5)? Could you provide some information and explanation?

Are the difference in age groups comparable with Spanish population?

Additionally, exploring the dietary patterns among the participants would have been valuable, given the crucial role of diet in influencing changes in body composition. This insight could have enhanced the identification of correlations between dietary habits and the observed physical variations within the study sample. Moreover, Spanish have a Mediterranean diet that can help to reduce age effect on body. This point need to be reported in the result section or addressed as limit in the discussion.

Could you add some information about alcohol intake?

Page 8 section “Body composition changes stratifying by sex according to changes in OPA” is really hard to read, please improve it reporting the majority of these results in a table that can help the reader.

Considering citation [5] at line 59, it would be advisable to cite a more recent study (including a systematic review) due to the numerous emerging studies that examine the correlation between sarcopenia and aging.

Please is important to include a citation at line 316 “Several studies have previously documented changes in body composition during aging; however, those with larger sample sizes were conducted in Asian populations or had a shorter duration.”

Minor comments a major English revision is required, following some examples:

L 34 – Please only use the abbrevation “OPA”

L 51 Please consider making a grammar revision: “The older population continues to grow”

L61: Please consider making a revision of “All these changes lead to suffer three important physio pathologies which also increase the risk of suffering frailty  and dependence during the aging process: sarcopenia, obesity and osteoporosis, which could be developed simultaneously”

I suggest you to use “All these changes contribute to the development of three significant physiopathologies,which […]”

L 75 - Please only use the abbrevation “OPA”

L 83-84: “which are the characteristics that makes effective these activities.” Please correct with: “characteristics that make these activitis effective.”

L 102: please remove activities after PA, as it is already present in the abbreviation PA

L 103: please remove as in the sentence: “Thus, most of the participants could be considered as active.”

L 105: please use the past tense: “agreed

L 148: please use the present tense: “spend” in the question: “How much time do you spent in them per week?”

L 152: please use “were” in the sentence: “Sedentary and walking time was recorded “

L 153: please use “walk” or “spend walking” in the question: “How many hours do you usually walking per day?”.

L225: please use “increase” instead of “increased”.

L 245: “in the Youngest group”

L 308: please use “lies” instead of “lie”

L 309: please add “the” before “main findings”

L 331: please use “continues” instead of “continue”

L 311: please use “there was” instead of “there is”

L 312: please revise the sentence “while in female seems not to affect.” I think it’s better using: “…while in females, it appears not to have an effect."

L 325: please make a grammar revision of this sentence: “In line, and combination with this, as changes in height occur during ageing, it could be affecting also BMI values and changings making it difficult to elucidate whether the differences or the absence of them are really due tissue-related changes during ageing”.

L 339: please add “a” before “previous study”

L 339: please consider to use “male” instead of “men”

L 341: please consider making a revision of “For Japanese women, it was described an increase in FM till their fifties and then, they seem to maintain it”. I suggest: “For Japanese women an increase in FM until their fifties was described, after which they seemed to maintain it…”

L 349: please consider making a revision of “Such increases in waist and hip indicates a higher risk of mortality …”  I suggest you to write “ These increments in waist and hip measurements indicate …”

L 367: please add “a” before “lighter”.

L 394: please use the past tense “there was” instead of “there is”

L 395-396: please consider making a revision of this sentence to improve clarity: “However, males who stopped OPA decrease their FFMI happened in males while it has no effect in females”. I suggest you to write: “However, males who discontinued OPA experienced a decrease in their FFMI, an effect not seen in females.”

Reviewer 2 Report

Comments and Suggestions for Authors

The  paper is  interesting  and the investigation is well  conducted . The  sample  is large and followed  for a long  time . The analysis  is  important for the population studied  and  the  clinical implication of  the subject. 

 Some  parts need  to  be  clarifyed . In the method session the authors sould clarify why they did not use the  IPAQ questionnaire  al least  in the initial phase  and at  the  end to identify  the  modifications of the intensity of the  exercise  and therefore  to  ditinguish  the different range of physical  activity practiced , Did you measure  in alternative  the daily steps, normally estimated by a simple app in the  mobil phone? How did you measure the body composition? By BIVA analysis? . 

Author Response

First of all, we would like to thank reviewers for the time devoted for this constructive review, for they positive comments and for giving us the opportunity to resubmit the manuscript. We believe that the manuscript has been improved accordingly and it is strong enough after this important review process.

We have taken into account every single comment and here we attach a point-by-point response to all of them; accordingly, we have also highlighted any changes made to the manuscript in yellow.

Comment 1: In the method session the authors sould clarify why they did not use the IPAQ questionnaire at least in the initial phase and at the end to identify the modifications of the intensity of the exercise and therefore to ditinguish the different range of physical activity practiced, Did you measure in alternative the daily steps, normally estimated by a simple app in the mobil phone?

Response 1: The reviewer is correct; including the IPAQ questionnaire would have enhanced the quality of the study. However, since it was not included in the baseline assessment, the decision was made not to include it in the follow up one either, and therefore, we lack data for this sample and its intensity. This is undoubtedly a limitation that have been now reported in that section(line 372). Data about activity was measured only through questionnaires about walking hours and the hours spent participating in organized activities.

Comment 2: How did you measure the body composition? By BIVA analysis?. 

The body composition measurements were performed using estimated data generated by the models developed by the TANITA brand, which are widely utilized. However, the older model, TANITA BC-418MA, used in this study, lacks the capability to collect vector data. Consequently, novel BIVA analyses of this type could not be performed.

Reviewer 3 Report

Comments and Suggestions for Authors

The paper, titled "Longitudinal changes in body composition of non-institutionalized Spanish older adults after 8 years of follow-up: Effects of sex, age, and organized physical activity," aims to assess the evolution of body composition in Spanish older adults and investigate the influence of organized physical activities on these changes. The paper is interesting and has merit, however, I have a few comments.

1.     Figure 1 is a cake diagram, such types of diagrams are not appropriate for a manuscript, please change it to a more suitable type of graph or a table.

2.     Results in lines 255-298 is very dense and hard to follow up, I suggest to present it in a table. Also make sure figures 2 and 3 do not duplicate data from the text or tables.

3.     Have you also performed analysis based on a type of physical activity or intensity of physical activity?

4.     Was data about other factors that could influence the body composition (diet, socioeconomic status, etc.) also acquired?

5.     One limitation of study is also a significant difference in included male and female participants.

Round 2

Reviewer 1 Report

Comments and Suggestions for Authors

NA

Reviewer 2 Report

Comments and Suggestions for Authors

The paper has been modifyed following the suggestions.Some aspects have been improved .